# Aftereffects to Prism Exposure without Adaptation: A Single Case Study

**DOI:** 10.3390/brainsci12040480

**Published:** 2022-04-05

**Authors:** Federica Albini, Alberto Pisoni, Anna Salvatore, Elena Calzolari, Carlotta Casati, Stefania Bianchi Marzoli, Andrea Falini, Sofia Allegra Crespi, Claudia Godi, Antonella Castellano, Nadia Bolognini, Giuseppe Vallar

**Affiliations:** 1Department of Psychology, University of Milano-Bicocca, 20126 Milano, Italy; alberto.pisoni@unimib.it (A.P.); a.salvatore3@campus.unimib.it (A.S.); nadia.bolognini@unimib.it (N.B.); 2Neuro-Otology Unit, Division of Brain Sciences, Imperial College London, London SW7 2AZ, UK; elenacalzolari@hotmail.com; 3Experimental Laboratory of Research in Clinical Neuropsychology, IRCCS Istituto Auxologico Italiano, 20155 Milano, Italy; c.casati@auxologico.it; 4Department of Neurorehabilitation Sciences, IRCCS Istituto Auxologico Italiano, 20155 Milano, Italy; 5Laboratory of Neuro-Ophthalmology and Ocular Electrophysiology, IRCCS Istituto Auxologico Italiano, 20155 Milano, Italy; s.b.marzoli@auxologico.it; 6Neuroradiology Unit and CERMAC, IRCCS San Raffaele Scientific Institute, Vita-Salute San Raffaele University, 20132 Milano, Italy; falini.andrea@unisr.it (A.F.); sofia.crespi@hsr.it (S.A.C.); claudia.godi@hsr.it (C.G.); castellano.antonella@hsr.it (A.C.)

**Keywords:** prism adaptation, hemianopsia, multisensory integration, single case study

## Abstract

Visuo-motor adaptation to optical prisms (*Prism Adaptation,* PA), displacing the visual scene laterally, is a behavioral method used for the experimental investigation of visuomotor plasticity, and, in clinical settings, for temporarily ameliorating and rehabilitating unilateral spatial neglect. This study investigated the building up of PA, and the presence of the typically occurring subsequent *Aftereffects* (AEs) in a brain-damaged patient (TMA), suffering from apperceptive agnosia and a right visual half-field defect, with bilateral atrophy of the parieto-occipital cortices, regions involved in PA and AEs. Base-Right prisms and control neutral lenses were used. PA was achieved by repeated pointing movements toward three types of stimuli: visual, auditory, and bimodal audio-visual. The presence and the magnitude of AEs were assessed by proprioceptive, visual, visuo-proprioceptive, and auditory-proprioceptive straight-ahead pointing tasks. The patient’s brain connectivity was investigated by Diffusion Tensor Imaging (DTI). Unlike control participants, TMA did not show any adaptation to prism exposure, but her AEs were largely preserved. These findings indicate that AEs may occur even in the absence of PA, as indexed by the reduction of the pointing error, showing a dissociation between the classical measures of PA and AEs. In the PA process, error reduction, and its feedback, may be less central to the building up of AEs, than the sensorimotor pointing activity per se.

## 1. Introduction

Multisensory integration is a mechanism for maximizing sensitivity to sensory events, which enhances perception and attention, encompassing the improvement of visual detection [1,2], visual and auditory localization [3,4], and the reduction of saccadic latencies [5,6,7,8]. In stroke right-brain-damaged patients with left *Unilateral Spatial Neglect* (USN) [9,10,11], and/or with homonymous hemianopia [12], multisensory integration improves the report of visual events occurring in the side of space or in the visual half-field contralateral to the side of the lesion [13,14,15,16]. Multisensory integration has also been used to enhance the effects of visuo-motor adaptation to optical prisms, *Prism Adaptation* (PA) [17]. Finally, recent evidence indicates that, in a rehabilitation setting, intensive audio-visual multisensory stimulation for two weeks brings about an improvement of symptoms of USN, overall comparable to that induced by the PA method [18].

PA has long been used for investigating visuomotor plasticity in healthy participants [19], as well as, in clinical settings, for ameliorating disorders of spatial representation and attention, such as USN [20,21,22,23]. In a standard PA paradigm, participants are required to make visuo-motor pointing movements toward a visual target, while they are wearing prismatic goggles displacing the visual field. The displacement brings about a misalignment between the visual and the proprioceptive spatial cerebral maps, resulting in a pointing error towards the side of the visual displacement. Throughout the repeated pointing movements, the error diminishes, due to an adaptive realignment of the eye-hand coordinates, involving sensorimotor brain plasticity. Specifically, during prism exposure, errors in the pointing task rapidly decrease for the most part during the first 15 trials, approaching the participant’s accuracy before exposure to prisms [19]. A model of PA [24,25] posits that two different mechanisms occur, to achieve a stable adaptation: (a) *recalibration,* which takes place in the first few pointing trials, and has strategic-cognitive features; (b) *realignment*, which consists in an automatic and implicit reorganization of the spatial map, building up with the repetition of pointing. The reduction of the pointing error in the first trials appears to depend on the process of recalibration. Once the prisms are removed, the participants’ pointing movements are still deviated, but in the opposite direction with respect to the previous visual displacement: this deviation after PA is termed *Aftereffects* (AEs). AEs in the direction opposite to that of the prism-induced displacement depend on both recalibration and realignment, and, therefore, several pointing trials, adequate for both processes to occur, are appropriate [26]. AEs, which are the main index that PA has occurred, are observed in the sensorimotor and visual domains [19,26]. AEs may be assessed through straight ahead pointing movements without visual control (*Proprioceptive Test*), and pointing to a position on a horizontal plane, towards a point corresponding to the projection of a visual target, centered on the mid-sagittal plane of the participants’ body (*Visuo-proprioceptive Test*). The proprioceptive and the visuo-proprioceptive AEs consist in a deviation of pointing in a lateral direction, with respect to the target straight ahead, which is opposite to the direction of the prism-induced deviation. In a third task (*Visual Test*) assessing AEs, participants communicate through a verbal response to the examiner when the perceived position of a visual target moving horizontally is aligned to the mid-sagittal plane of their trunk. Unlike the proprioceptive and visuo-proprioceptive AEs, AEs measured by the visual test occur in the same direction of the optical displacement brought about by the prisms [19]. PA may also induce cross-modal phenomena, such as modulating the perception of acoustic space: adaptation to prisms that displace laterally the visual field produces a corresponding shift in the subsequent localization of a sound [27,28,29]. In a study in healthy participants [30], using the standard pointing task to visual targets for assessing PA, in the condition with no auditory signals, primarily proprioceptive AEs were found; in the condition with pointing paced by auditory signals, with participants being instructed to carefully synchronize their movements to the auditory signal, also visual AEs occurred; in the condition with no synchronization between auditory signals and pointing, AEs were primarily visual. These findings suggest both an overall alerting effect of auditory signals (condition with instructions to synchronize) and bimodal sensory integration (condition with no instructions to synchronize). Visual and proprioceptive AEs occur also after PA to auditory, and not only to visual targets. Pointing to bimodal audio-visual targets enhances PA, with participants being more precise and faster in error correction when the target combines the signal from the two sensory modalities, speeding up the adaptation process. As to the neural correlates of these effects, the posterior parietal cortex (PPC), and occipital and temporal regions are involved in both PA and visuo-acoustic integration [31,32]. Specifically, during PA, in the earliest phase of prism exposure, the anterior intraparietal sulcus is primarily implicated in error detection, and the parieto-occipital sulcus mainly in error correction. Conversely, neural activity in the cerebellum progressively increases during PA, suggesting a key role of this region for spatial realignment. The cerebellum might promote changes of neural activity in the superior temporal cortex, activated during the later phase of PA [32]. These regions may then mediate the AEs, and the effects of PA on cognitive spatial representations, as found in patients with USN [33]. This evidence may be considered in the context of the two visual systems model, distinguishing “vision for perception” and “vision for action” [34,35]. Although the two systems are highly interactive, the ventral stream (“vision for perception”), passing from the primary visual cortex (V1) through to the inferior parts of the temporal lobe, is considered to mediate the transformation of the contents of the visual signal into representations that guide conscious perception, recognition, and memory for objects. In contrast, the dorsal stream, passing from V1 through to various areas in the PPC, is generally considered to mediate the visual guidance of action, primarily in real time [36].

In sum, there is evidence that the PA process may have multisensory properties, able to extend the adaptation to the optical deviation induced by prisms to acoustic or visuo-acoustic targets, in addition to the typically used visual targets.

The present single-case study investigated the effects of PA, and the presence of the subsequent aftereffects (AEs), using target stimuli for pointing in three different modalities: visual, auditory, and bimodal audio-visual. The PA procedure was performed by healthy controls and a patient (TMA) with apperceptive agnosia [37], and a right visual half-field deficit, due to a bilateral atrophy of the parieto-occipital cortices. The aim of the study was to assess whether PA and AEs may, and to which extent, or may not occur, with visual, auditory, and visuo-auditory targets in a patient with damage to regions involved in these processes.

## 2. Materials and Methods

### 2.1. Participants

#### 2.1.1. Patient TMA

Patient TMA was a right-handed [38] 67-year-old woman with 18 years of education. In April 2015 the patient was referred to the IRCCS Istituto Auxologico Italiano for a neuropsychological assessment. Three years before (July 2012), TMA had developed a sudden loss of vision. The patient reported to be able to see sometimes only flashes of light; she also reported the occurrence of visual hallucinations. Gradually, TMA regained her sight, but with persistent difficulties in the discrimination of objects, faces, and color. The patient suffered a low-grade non-Hodgkin lymphoma with a monoclonal peak IgA (immunoglobulin A), IVA stage [39]. The patient had been treated with an R-CVP (Cyclophosphamide-Prednisolone-Rituximab-Vincristine) protocol for a non-Hodgkin lymphoma [39]. TMA, during the third cycle of this therapy, developed a leukoencephalopathy, presumably as a reaction to Rituximab [40]. A neurological examination (October 2012) showed a right homonymous hemianopia, of which the patient was fully aware [41]. An ophthalmoscopic exam revealed papillary margins preserved bilaterally. Humphrey visual field perimetry (HVF 30.2) confirmed the right homonymous hemianopia with no macular sparing (Figure 1). The results of the baseline neuropsychological assessment are summarized in Table A1 in Appendix A.

Magnetic Resonance Imaging (MRI) was performed on a 3.0T Ingenia CX scanner (Philips Healthcare, Best, The Netherlands) at the Neuroradiology Unit and CERMAC of the IRCCS Ospedale San Raffaele (Milan, Italy). The exam included conventional MRI, as well as Diffusion Tensor Imaging (DTI) data acquired with HARDI (high angular resolution diffusion-weighted imaging, 60 gradient directions, b-value = 3000 s/mm^2^). Conventional MRI (last in March 2014) showed outcomes of probable gliotic malacic-nature with bilateral atrophy of the parieto-occipital lobes, and small and multiform vascular lesions in the white matter of the frontal lobes bilaterally (Figure 2).

MR tractography reconstruction of the main white matter brain pathways was performed with a probabilistic q-ball algorithm, according to the pipeline described in Pieri et al. [42]. Tractography analysis showed an asymmetry of the representation of the superior longitudinal fasciculi, with right temporo-parietal connections less represented in the right fascicle. A reduction of the representation of the posterior and dorsal fibers of the inferior longitudinal fasciculus (ILF), and of the inferior fronto-occipital fasciculus (IFOF), was also observed in the left hemisphere, possibly related to the gliotic lesions visible on conventional MRI (Figure 2). Left optic radiation was damaged as well, with a reduction of fractional anisotropy along its fibers, as shown in Figure 3.

A neuropsychological assessment was performed in March and April 2015. As for visual perception, TMA was unable to perform the Visual Object and Space Perception Battery (V.O.S.P.) [43], since she did not pass the Screening Test, showing a severe deficit in the discrimination of figures from the background. TMA’s performance in the Line Orientation Test [44] was also defective. The Birmingham Object Recognition Battery (B.O.R.B.) [45] was not administered, as TMA failed in both the Discriminating Figures and the Reading Letters subtests. As for linguistic abilities, spontaneous speech, and phonemic and semantic fluencies [46] were preserved. At the Neuropsychological Exam for Aphasia (Esame Neuropsicologico per l’Afasia, E.N.P.A.) [47], writing of non-words was defective; the E.N.P.A.’s reading test (both words and non-words) and the Token Test [48] could not be performed, due to the patient’s misrecognition of letters and forms. Performance at the Boston Naming Test [49] was impaired since the patient was unable to recognize most of the proposed images (total score: 13 correct responses out of 60 pictures). TMA showed no deficit of auditory verbal memory, both in the short- and in the long-term, and no deficit in short-term visuo-spatial memory [50,51,52]. As for visuo-spatial exploration, in Albert’s line cancellation task [53], TMA made omission errors in the right-hand side of the sheet. The H letter cancellation task [54,55] was not performed since TMA was unable to discriminate among letters and to recognize them. As for drawing by copy and from memory, TMA’s performance was defective in copy drawing, with the model being reproduced in a fragmented way. Conversely, drawing from memory was preserved. Overall, these findings are consistent with the conclusion that TMA’s visual recognition deficit may be traced back to apperceptive visual agnosia [37]. A comparatively better performance in drawing from memory than in copy drawing has been previously reported in visual form agnosic patient DF [56].

#### 2.1.2. Control Participants

Twenty-four healthy right-handed [38] participants (12 males, mean age = 63.12 years, SD = ±5.227, range = 54–74; mean educational level = 10.16 ± 3.225 years, range = 5–18), with no history and evidence of neurological and psychiatric disorders, and a normal or corrected-to-normal vision, participated in the study.

### 2.2. Procedure

Participants underwent five experimental sessions in five different days, with an inter-session interval of at least 24 h (range = 24–96). Using a procedure adapted from Calzolari et al. [17], each PA session lasted about one hour and included: (a) a pre-exposure phase, (b) a PA phase and (c) a post-exposure phase, identical to the pre-exposure phase.

On the first day of testing, before the actual session, all participants performed an auditory localization test, as a screening evaluation, to exclude hearing loss and auditory localization deficits. During the screening evaluation participants achieved 24 manual pointing movements with the right hand towards auditory targets presented in a pseudorandom fixed order, at four different locations (+10°, +20° rightwards, and −10° and −20° leftwards, with respect to the participant’s body midline, see Figure 4A). None of participants committed errors in this screening auditory test.

#### 2.2.1. Prism Adaptation EXPOSURE phase

In the first session, participants adapted to a 10° leftward visual shift, induced by 20-dioptre, base-right prism glasses (Bernell Corporation, Mishawaka, IN, 46545, US). Visuo-motor adaptation was achieved using the right hand, with participants performing 24 manual pointing movements towards a visual target, namely: a red LED presented for 150 ms at four different locations (+10° and +20° rightwards, and −10° and −20° leftwards, with respect to the participant’s body midline). Visual targets were presented in a pseudorandom fixed order.

In the other four sessions, visuo-motor adaptation was achieved by the execution of 92 manual pointing movements towards a target presented at three different positions (+10°, rightwards, and −10° and −20° leftwards, with respect to the participant’s body midline) in a pseudorandom fixed order, which was the same in all the four sessions and for all participants. TMA, due to her right hemianopia (Figure 1) was unable to detect and report the +20° rightward target. Accordingly, this target was not used in the experimental study (Figure 4A). The four sessions differed in two respects: (a) the type of goggles worn during the adaptation phase (20-dioptre, base-right prism glasses, inducing a 10° leftward visual shift, or neutral/sham lenses, inducing no deviation of the visual field); (b) the modality of the targets in the adaptation phase: auditory (white noise burst), or audio-visual, with a simultaneous presentation of the LED and of the white noise burst. Thus, the four session conditions were: prism/auditory, prism/audio-visual, neutral/auditory, neutral/audio-visual. Both TMA and control participants performed the five experimental sessions in the following order (Figure 5): (a) PA to 24 visual targets, (b) PA to 92 audio-visual targets, (c) sham PA to 92 audio-visual targets, (d) sham PA to 92 auditory targets, and (e) PA to 92 auditory targets. The apparatus for the presentation of the targets was the same used by Calzolari et al. [17] and is shown in Figure 4A. The temporal sequence of the real PA and sham PA conditions administered to TMA is shown in Figure 5. Participants were tested individually in a silent room.

#### 2.2.2. Pre- and Post-Exposure Phases

In line with previous studies [17], to investigate whether PA had been successful in bringing about the complete AEs, the presence and the magnitude of the proprioceptive, audio-proprioceptive, visuo-proprioceptive, and visual AEs was assessed before and after each adaptation condition (see Figure 5). The aim of this procedure was to detect possible dissociations among AEs. A previous study in patient MM, with bilateral occipital and left cerebellar damage, had shown the expected preserved reduction of the pointing errors to rightward displacing prisms, but not the leftward AEs in the proprioceptive straight-ahead task; conversely, the visuo-proprioceptive and the visual AEs were preserved [17]. AEs were assessed in the following order:*Proprioceptive. P*articipants, seated in front of a table, with eyes closed, received instructions to point with the right index finger to the location on the table surface, perceived as the subjective straight-ahead (Figure 4B).*Audio-proprioceptive.* In darkness, participants received instructions to point with the right index finger to the location on the table surface subjectively perceived as the projection of a sound source. No information was given to participants about the location of the loudspeaker (Figure 4B).*Visuo-proprioceptive.* In darkness, participants received instructions to fixate a red LED placed in the straight-ahead position, at a 65 cm distance, and to point with the right index finger to the location on the table surface subjectively perceived as the projection of the light on the table. No information was given about the actual LED location, and a wooden box precluded participants from viewing the pointing movement, which then took place without any visual control (Figure 4B).*Visual.* In darkness, participants received instructions to stop verbally a red LED, moving horizontally just above eye level, at 65 cm from the participant’s mid-sagittal plane, when the light was subjectively perceived as straight-ahead. The 10 trials (five with the light moving from the right to the left visual periphery, five from the left to the right) were given in a random fixed order. For each test, 10 trials were given. Each participant performed the tests in the same order in the pre- and in the post-exposure phases. For each test, the mean deviation from the objective midline was calculated, both in the pre- and in the post- adaptation phase; positive values indicated a rightward deviation from the perceived body midline, negative values a leftward deviation [17].

### 2.3. Statistical Analyses

Firstly, in each session, the presence of PA and of AEs was assessed in the group of control participants. The presence of PA (i.e., the reduction of the initial pointing error in the exposure phase) was assessed via a series of linear mixed models [57], using the “lme4” R package [58] on the mean pointing deviation from the target in angle degrees. As independent fixed continuous variables, for auditory and audio-visual adaptation procedures, both the linear and the quadratic effects of the *Pointing number* (range = 1–92) were added to the model, as well as the *Lens type* (factorial, two levels: prismatic vs. neutral lenses). Concerning the visual adaptation procedure, a linear mixed model was computed with both the linear and the quadratic effect of the *Pointing number* (range = 1–24), as fixed effect, and the participants’ random intercept, since no neutral lenses were used in this condition. Likelihood ratio tests (LRT) were performed to explore whether the inclusion of the fixed independent variable *Pointing number* increased the model’s goodness of fit [59]. A by-subject random intercept was also added to account for inter-subject variability in the adaptation procedure.

For sessions #2 to #5, the presence of sensorimotor AEs in the proprioceptive, audio- proprioceptive, and visuo-proprioceptive straight-ahead tests, was assessed with a repeated-measures ANOVA on the participants’ mean straight-ahead scores, with the within-subjects main factors *Condition* (pre-exposure vs. post-exposure) and *Test* (proprioceptive, audio-proprioceptive, and visuo-proprioceptive AEs). Only for session #1 (i.e., 24 visual targets during the adaptation phase, and inclusion of the visual straight-ahead test scores in the pre- and in the post-exposure to prisms assessments), a repeated-measures ANOVA on the participants’ mean straight-ahead deviations in the proprioceptive, audio-proprioceptive, visuo-proprioceptive and visual straight-ahead tests was performed, with the within-subjects main factors *Condition* (pre-exposure vs. post-exposure), and *Test* (proprioceptive, audio-proprioceptive, visuo-proprioceptive and visual straight-ahead tests).

Secondly, the difference in the adaptation trends between control participants and patient TMA was assessed. Linear mixed effects models [60], using the “lme4” R package [58], were applied, considering the pointing deviations from the target position in angle degrees as a continuous dependent variable for each adaptation condition. As independent continuous variables, both the linear and the quadratic effect of *Pointing number* were added to the model (range 1–24 for the visual and 1–92 for the auditory and bimodal audio-visual prismatic procedures), as well as of *Group* (factorial, two levels: patient vs. control participants). A linear mixed effects model [60] was used to compare the performances of patient TMA with those of control participants. LRT were performed to explore whether the inclusion of the fixed independent variable *Pointing number* increased the model’s goodness of fit [55]. A by-subject random intercept was also added to account for inter-subject variability in the adaptation procedure. The summary of the final model is reported, together with significance level based on Satterthwaite’s degrees of freedom approximation in the “lmerTest” R Package [61] Adaptation trends were also explored via a series of segmented regression analyses, using the “segmented” package in R [62,63]. This analysis estimates if there are different phases in the rate of decrease of the shifting errors at some point during the procedure. Bilinear regressions were fitted using an iterative parameter estimation procedure, using as starting point for breakpoint estimation the putative number of trials when recalibration and realignment should take place (i.e., 10 and 20 pointing movements) [64]. This procedure was performed for healthy controls in the three adaptation procedures, while it could not be carried out for the patient, since she represented a sample of only one observation. Estimates for the breakpoints are provided, as well as the slope values for the computed segments.

Finally, the presence of AEs in TMA was assessed: TMA’ performances in each straight-ahead test were compared to those of control participants, with t-tests for case-control comparisons [63].

Statistical analyses were carried out with SPSS (IBM SPSS Statistic version 21, IBM Corp., Armonk, NY, USA) and with the software R (R Development Core Team, 2014, https://www.r-project.org/, accessed on 6 March 2022). In all ANOVAs, significant effects and interactions were explored with Bonferroni post-hoc test for multiple comparisons. To determine effect sizes, partial eta squared (ηp2) values for F-tests [61] are provided.

## 3. Results

### 3.1. Adaptation during the Exposure Phase

Statistical analyses on the adaptation phase show that control participants reduce their initial pointing error in the visual procedure, as the quadratic effect of pointing number is significant (b = −24.02; t_550_ = −9.93; *p* < 0.001; see Figure 6 for full results).

Concerning the comparison between prismatic and neutral lenses in the audio-visual and in the auditory procedures, results differ. As for the audio-visual procedure, the final model includes the main effects of *Pointing trial*
*number* [χ^2^(2) = 514.02, *p* < 0.001] and of *Lens type* [χ^2^(1) = 29.76, *p* < 0.001], as well as their interaction [χ^2^(2) = 623.027 *p* < 0.001]. The model shows a significant interaction between the quadratic trend of *Pointing trial number* and of *Lens type* (b = 35.53, t_4364_ = 15.36, *p* < 0.001), with prismatic lenses showing a quadratic decreasing trend, while the pointing shift with neutral lenses has a linear trend with lower errors since the beginning of the adaptation procedure (see Figure 7 and Table A2 in Appendix B, for full model results).

As for the auditory adaptation procedure, also in this case the final model includes the main effects of the quadratic trend of *Pointing trial number* [χ^2^(2) = 10.46, *p* = 0.005] and of *Lens type* [χ^2^(1) = 16.72, *p* < 0.001], as well as their interaction [χ^2^(2) = 14.3, *p* < 0.001]. However, the final model shows only a significant interaction between *Lens type* and the linear trend of *Pointing trial number* (b = −32.74, t_4364_ = −3.76, *p* < 0.001; see Table A3 in Appendix B, for full results). Inspecting the data, it appears that, while wearing neutral lenses, participants tend to produce no errors throughout the adaptation procedure; conversely, while wearing prismatic lenses, they tend to increase the rightward error during the end of the procedure (see Figure 8).

Concerning the comparison between the adaptation procedure with prismatic lenses between patient TMA and healthy control participants, for the visual adaptation procedure the final model includes the main effects of *Pointing number* [χ^2^(2) = 286.5, *p* < 0.001] and of *Group* [χ^2^ (1) = 4.3, *p* = 0.038], as well as their interaction (χ^2^(2) = 12.76 *p* = 0.002). Critically, the interaction between the quadratic trends for *Pointing trial number* and for *Group* is significant (b = 48.45, t_571_ = 3.37, *p* < 0.001), as control participants reduce their shift from the midline as the pointing procedure is performed, while the patient does not show this pattern, moving towards a greater rightward error and keeping a higher distance from the actual midline until the end of the procedure (see Figure 6 and Table A4 in Appendix B, for full results).

Similarly, for audio-visual adaptation, the final model includes the main effects of *Pointing trial number* [χ^2^(2) = 674.31, *p* < 0.001] and of *Group* [χ^2^(1) = 19.37, *p* < 0.001], as well as their interaction [χ^2^(2) = 17.71 *p* < 0.001]. The model shows a significant interaction between the quadratic trends of *Pointing trial number* and of *Group* (b = 29.08, t_2271_ =3.74, *p* < 0.001), since control participants gradually reduce their pointing error as the adaptation procedure is performed, while the patient gradually increases the rightward shift (see Figure 9A and Table A5 in Appendix B, for full results). The model of sham adaptation with audio-visual targets reports the main effects of *Pointing trial number* [χ^2^(2) = 9.37, *p* = 0.009] and of *Group* [χ^2^(1) = 18.83, *p* < 0.001], as well as their interaction [χ^2^(2) = 13.01, *p* = 0.001]. The final model shows a significant interaction between the linear trend of *Pointing trial number* and of *Group* (b = −13.21, t_2271_ = −3.09, *p* = 0.001): while healthy participants tend to keep their pointings around the midline, the patient shows rightward errors (see Figure 9B and Table A6 in Appendix B, for full results).

Finally, also for the auditory adaptation procedure, the final model includes the main effects of *Pointing trial number* [χ^2^(2) = 16.16, *p* < 0.001] and of *Group* [χ^2^(1) = 5.99, *p* = 0.014], as well as their interaction [χ^2^(2) = 6.43, *p* = 0.04]. The final model shows a significant interaction between the linear trends of *Pointing trial number* and of *Group* (b = 65.65, t_2271_ = 2.34, *p* = 0.019): control participants tend to keep their pointing errors around the midline, even though slightly shifted towards the right side of space, while TMA increases the shifting errors as the procedure is carried on (see Figure 10A and Table A7 in Appendix B for full results). Similarly, the model of sham adaptation with auditory targets reports the main effects of *Pointing trial number* [χ^2^(2) = 10.71, *p* = 0.004] and of *Group* [χ^2^(1) = 5.10, *p* = 0.02], as well as their interaction [χ^2^(2) = 7.87, *p* = 0.01]. The final model shows a significant interaction between the linear trend of *Pointing trial number* and of *Group* (b = 36.26, t_2271_ = 2.65, *p* = 0.008): the patient makes errors shifted rightwards with respect to the targets (see Figure 10B and Table A8 in Appendix B for full results).

Finally, the adaptation procedures of control participants were analyzed with segmented regressions, to assess if there are crucial points in the protocol where control participants or the patient modify their performance. In the visual procedure, control participants show two critical breaking points: at the 3.7 (SD = 0.59) and 14.05 (SD = 1.67) pointing trial number (see Figure 11A).

Concerning the bimodal audio-visual procedure, the performance slightly changes for control participants, as the first breakpoint overlaps with the one found for visual adaptation (mean = 4.99; SD = 0.26), while the second one is delayed at the 24.04 pointing trial number (SD = 2.36, see Figure 11B). It should be noted, however, that this procedure is longer than the visual one (92 vs. 24 pointing trials) and this may affect the breakpoint estimate (see Figure 11B).

Finally, for the auditory adaptation procedure both breakpoints are delayed, as compared with the audio-visual and visual procedures. The first breakpoint is only slightly postponed (mean = 6.55 pointing trial, SD = 0.7), the second one to a later phase of the adaptation procedure (mean = 33.1 pointing trial, SD = 3.5, see Figure 11C). It should be noted that these breakpoints values do not diverge from previous reports [65].

### 3.2. Aftereffects

#### 3.2.1. Control Participants

In the control group, after exposure to leftward deviating prisms, proprioceptive, audio-proprioceptive and visuo-proprioceptive straight-ahead AEs occur (Figure 12). Conversely, in the visual straight-ahead test no AEs are present. Repeated-measures ANOVAs show a significant main effects of *Condition* [24 visual targets: F_(1,23)_ = 44.65, *p* < 0.001; 92 auditory targets: F_(1,23)_ = 73.64, *p* < 0.001; 92 audio-visual targets: F_(1,23)_ = 82.54, *p* < 0.001], and of *Test* [24 visual targets: F_(2,46)_ = 6.80, *p* < 0.001; 92 auditory targets: F_(2,46)_ = 4.70, *p* = 0.014; 92 audio-visual targets: F_(2,46)_ = 5.78, *p* = 0.006], and a significant *Condition by Test* interaction [24 visual targets: F_(2,46)_ = 15.54, *p* < 0.001; 92 auditory targets: F_(2,46)_ = 3.99, *p* = 0.025; 92 audio-visual targets F_(2,46)_ = 5.34, *p* = 0.008]. Post-hoc comparisons for the interaction indicate that the mean straight-ahead post-adaptation deviation is shifted rightwards, as compared to the pre-adaptation deviation, in all tests (all *p* values ≤ 0.001), but the visual straight-ahead test (*p* = 0.097).

No AEs are present both in the sham prism adaptation condition with 92 audio-visual targets (*Condition by Test* interaction, 92 audio-visual targets F_(2,46)_ = 0.32, *p* = 0.722), and in the sham prism adaptation condition with 92 auditory targets, even if the interaction *Condition by test* is significant (F_(2,46)_ = 3.34, *p* = 0.044). Post-hoc comparisons show significant (*p* = 0.005) baseline differences only between the proprioceptive straight-ahead and the auditory-proprioceptive straight-ahead tests, and between the audio-proprioceptive straight-ahead and the visuo-proprioceptive straight-ahead tests (pre-exposure assessments: proprioceptive test: M ± SD = 0.15 ± 2.98; audio-proprioceptive test: M ± SD = −4.35 ± 5.92; visuo-proprioceptive test: M ± SD = −0.46 ± 2.40).

#### 3.2.2. Patient TMA


PA condition: *24 visual targets*. As compared to control participants, TMA shows AEs in the proprioceptive and audio-proprioceptive straight-ahead tests (Figure 13A). In the 24 visual target PA condition, Crawford-Howell t-tests for case-control comparisons show that the mean score of control participants (proprioceptive test: M = 2.25, SD = ±2.31; audio-proprioceptive test = 2.48 ± 3.34) does not differ from that of the patient in both the proprioceptive (5.05 ± 1.55, *p* = 0.24) and the audio-proprioceptive straight-ahead tests (−4.60 ± 15.55, *p* = 0.05). However, the mean score of control participants (visuo-proprioceptive test = 3.49 ± 1.62) differs from that of the patient in the visuo-proprioceptive straight-ahead test (−1.65 ± 7.43, *p* = 0.004): as compared with the control group, TMA’s mean straight-ahead post-adaptation deviation is not shifted rightwards.PA condition: 92 *auditory targets*. AEs in the visuo-proprioceptive straight-ahead tests are present (Figure 13B). A Crawford t-test shows that the mean score of control participants in the visuo-proprioceptive straight-ahead test (3.54 ± 1.85) differs from the patient’s score (14.5 ± 5.75, *p* < 0.001), namely: TMA shows larger AEs than control participants. The mean score of control participants (2.235 ± 1.55) differs from the patient’s score in the proprioceptive straight-ahead test (−1.20 ± 1.87, *p* = 0.040). Similarly, the mean score of control participants (2.17 ± 2.69 in the audio-proprioceptive straight-ahead test) differs from the patient’s score (−5.5 ± 2.91, *p* = 0.006). However, in these tests TMA’s mean straight-ahead post-adaptation deviation is not shifted rightwards, as expected after adaptation to prisms displacing the visual scene leftward.PA condition: 92 *audio-visual targets*. AEs in all straight-ahead tests are present (Figure 13C); t-tests for case-control comparisons show that the mean scores of control participants (proprioceptive test: 2.70 ± 2.22; audio-proprioceptive test: 2.30 ± 2.73; visuo-proprioceptive test: 4.05 ± 1.75) do not differ from those of TMA (proprioceptive test: 4.85 ± 1.61, *p* = 0.35; audio-proprioceptive test: 7.45 ± 3.19, *p* = 0.07; visuo-proprioceptive test: 4.05 ± 2.86, *p* = 1).PA condition: *sham.* Crawford t-tests show no significant differences between TMA’s mean straight-ahead post-PA deviations and those of control participants in all conditions (Figure 13D,E). 


TMA’s baseline pointing accuracy to visual targets was verified by analyzing her performance in the visuo-proprioceptive straight ahead pointing task. As compared to control participants, TMA shows no differences in all five days (Figure 14). Crawford-Howell t-tests for case-control comparisons show that the mean scores of control participants (first day: M = −1.31, SD ± 2.37; second day: M = −0.68, SD ± 2.66; third day: M = −1.12, SD ± 2.24; fourth day: M = −0.46, SD ± 2.40; fifth day: M = 0.07, SD ± 2.58;) do not differ from the patient’s scores (M = −1.2, SD = ±5.89; M = −2.65, SD ± 3.26; M = 1.85, SD ± 2.09; M = 0.95, SD ± 0.64; f M = −2.8 SD ± 3.32; all *p* > 0.05) (*p* = 0.96; *p* = 0.47; *p* = 0.20; *p* = 0.84: *p* = 0.29).

### 3.3. Summary of Statistical Results

Control participants, during real PA (i.e., when wearing prismatic lenses shifting the visual scene leftward), show the expected trend of adaptation in all target conditions (visual, auditory, and audio-visual), namely: an initial leftward pointing error, which decreases throughout the exposure phase. For the auditory PA procedure, control participants appear to overcompensate as the test is performed, resulting in a small rightward shift at the end of the protocol (Figure 8). This trend may be taken as evidence of the occurrence of recalibration and realignment, namely: the two-processes involved in PA [19]. Instead, as compared to control participants, TMA shows a larger error variability during the adaptation phase. In the 24 target visual PA condition, control participants, after the exposure to leftward deviating prisms, exhibit all rightward AEs, but the visual ones. TMA too shows AEs, with this pattern: (a) after PA with 24 visual targets, significant rightward AEs in the proprioceptive and in the audio-proprioceptive straight-ahead tests; (b) after PA with 92 audio-visual targets, significant rightward AEs in all straight-ahead tests (proprioceptive, audio-proprioceptive and visuo-proprioceptive); (c) after PA with 92 auditory targets, significant rightward AEs in the visuo-proprioceptive straight-ahead test.

## 4. Discussion

In this study the building up of PA to a leftward prism-induced shift of the visual scene, and the presence of the subsequent AEs, were assessed in a brain-damaged patient (TMA) with apperceptive agnosia and a right visual half-field deficit, showing a bilateral atrophy of the parieto-occipital cortices, regions involved in both the PA and AEs phenomena [32]. Prior to the discussion of the pattern of PA and AEs shown by patient TMA, the possibility should be considered that the patient also suffered from a primary deficit of reaching and grasping such as in optic ataxia [66,67,68], In reaching, the main pattern of errors of optic ataxia includes dysmetria, with hypermetric and hypometric errors, rather than a bias towards an hemianopic half-field, which is very infrequently reported in these patients [69], since a main feature of optic ataxia is that it cannot be traced back to primary peripheral sensorimotor deficits. Patient TMA shows accurate pointing in the baseline visuo-proprioceptive test (Figure 14). Her pointing errors in the adaptation tasks, both with real and sham prisms, showed a rightward deviation, towards the hemianopic half-field. This contralesional deviation may be interpreted as a compensatory behavior, as the contralesional pointing and reaching errors made by hemianopic patients in line bisection [70,71]. Be as it may, however, the putative and unlikely presence of optic ataxia does not appear to have prevented the building up of AEs in the absence of a detectable PA.

One may also wonder, since the patient was assessed three years after the onset of the brain lesion, whether the AEs found in the present study were based on functional recovery, due to plastic neural changes, that is, whether they were or not present in an early period after lesion onset. A definite response to this hypothesis cannot be provided. It should be noted, however, that such a putative recovery of AEs is not the result of specific learning processes, promoted by training [72] since TMA’s performance for PA and AEs was assessed for this first time in the present study.

### 4.1. Aftereffects (AEs) without Error Reduction

TMA’s trend of error reduction in the adaptation phase and the subsequent AEs were compared to those of control participants, matched for age and educational level. The trend analysis on error reduction in the adaptation phase shows that control participants reduce their initial pointing error in all PA conditions (visual, audio-visual and auditory), with a slight overcompensation in the auditory PA protocol. This trend of PA is in line with previous reports [64]. Concerning the comparison between the patient and the control group, TMA shows a greater error variability during the PA procedures, with a reduced, if not absent, error reduction and, accordingly, no adaptation to the prismatic shift. This result could be ascribed, at least in part, to the patient’s severe disorders in the visual domain (see case report, Section 2.1.1). Critically, as for TMA’s performance in the auditory PA procedure, it is well known that precisely localizing an acoustic stimulus without the aid of other sensory inputs is much more difficult than localizing a visual stimulus [65,66]. Furthermore, PA affects the perception of acoustic space, bringing about a deviation in the acoustic localization of a sound [73,74]. Adaptation to prisms displacing the visual scene, using acoustic stimuli for pointing, may interact with the general difficulty in precisely identifying the source of a sound, accounting for TMA’s error variability in her pointing to acoustic stimuli. This is supported also by the pattern of performance of control participants in this protocol, which is less accurate than those found in the visual and audio-visual procedures. Although patient TMA shows AEs in all conditions of PA, all three AEs (audio-proprioceptive, visuo-proprioceptive and proprioceptive) are present only with PA to audio-visual stimuli (Figure 13C); for PA with visual stimuli the proprioceptive and audio-proprioceptive, but not the visuo-proprioceptive, AEs are present (Figure 13A); for PA with auditory stimuli only visuo-proprioceptive AEs were found (Figure 13B). The presence of complete AEs only with PA to audio-visual targets, as compared to unimodal (visual or auditory targets) indicates that multisensory stimuli, used as targets for the pointing task, may enhance the development of AEs after exposure to prisms displacing the visual scene. More generally, these findings are in line with the wide evidence indicating that multisensory stimulation enhances behavioral performance, for instance in detection tasks [73,74]. The present findings appear to extend this enhancement to the development of AEs after PA.

### 4.2. Recalibration and Spatial Realignment as Indipendent Mechanisms of PA

In detail, according to current accounts of the mechanisms of PA [19], two main processes occur throughout the PA procedure: recalibration and spatial realignment. Recalibration is the top–down, strategic, and voluntary component of PA, which brings about a quick and major reduction of the pointing error during the initial trials. Spatial realignment is conceived as a bottom-up, automatic, and unaware form of perceptual learning and is responsible for the AEs after PA; spatial realignment builds up in later pointing trials [19]. Figure 6, Figure 7, Figure 8, Figure 9, Figure 10 and Figure 11 show the early rapid major reduction of the pointing error in control participants, and the absence of this pattern in TMA. The significant AEs shown by the patient suggest that spatial realignment is largely preserved. Both the patient and control participants exhibit significant rightward AEs in the proprioceptive straight-ahead test, namely the more sensitive index that PA has occurred [75], after the visual and audio-visual procedures used in this study to detect the presence of PA. In line with these findings, both TMA and control participants show no AEs in the PA sham condition with neutral lenses.

The present findings in patient TMA also suggest that the aware recalibration process, as indexed by the absence of reduction of the pointing error, particularly during the early trials of the adaptation phase, may be not a crucial component for AEs to occur. A recent study in healthy participants [76], combining virtual reality and haptic robotics, shows that AEs in a direction opposite to that of the shift may occur under conditions in which patients are largely unaware of the visuo-motor mismatch between their hand and the virtual hand’s position. Specifically, during training, a rotational shift was induced between the position of the participant’s real hand and that of the virtual one, to trigger sensorimotor recalibration Furthermore, there is evidence that AEs after PA may occur also when right brain-damaged patients with USN wear prisms displacing the visual scene rightward, while performing visuo-motor ecological activities (e.g., opening and closing jars with the corresponding lids, sorting and playing cards, assembling jigsaw puzzles). Under these conditions, participants are largely unaware of the “direct effect” (i.e., the pointing error in the direction of the prism-induced displacement of the visual scene) of prism exposure [22,23]. The ecological tasks do not feature any pointing, with an apparent mismatch in the early pointing trials (of which participants are aware) between the intended pointing direction and the position of the target in front of them, involving instead the manipulation of objects, with no apparent and direction-specific lateral shift of the visual scene, as indexed by the pointing error. Finally, healthy participants, when PA is achieved by multiple-step (an unaware condition, because of the progressive stepwise increase from 2 to 10 deg of optical deviation) exposure to wedge prisms, show larger AEs, and transfer to the non-exposed hand for visual and auditory pointing tasks, as compared to the classical single-step condition. The latter condition (single-step), at variance from the former one (multiple-step), features awareness of the prism-induced shift, since participants are directly exposed to the full 10 deg deviation throughout the whole PA process [77]. The pattern of unawareness of the optical deviation and of increased AEs after the multiple-step PA in the unaware condition mimics the behavior of patients with left USN [21]. 

### 4.3. Anatomo-Functional Neural Correlates Involved in PA

The recalibration and spatial realignment processes have different anatomo-functional correlates, with spatial recalibration being supported by a network comprising the PPC, and realignment more related to the cerebellum [33,78,79]. As for functional connectivity, a recent meta-analysis of functional Magnetic Resonance Imaging (fMRI) studies shows in healthy adults an association between PA (in most studies to a prism-induced rightward shift) and activity in PPC and cerebellar clusters, with reduced bilateral parieto-frontal connectivity, and increased fronto-limbic and sensorimotor network connectivity. Conversely, in right brain-damaged patients with left USN different circuits are found, including an activity cluster in the intact left occipital cortex, consistent with some shift towards a role of the left hemisphere in spatial processing [80]. In one fMRI study [81] resting-state functional connectivity (RSFC) in healthy individuals was measured before and after PA, which modulates functional connectivity of the intraparietal sulcus of the PPC with several brain networks. When contrasted rightward vs. leftward PA, a reduction of RSFC in the “spatial navigation network” (right PPC, hippocampus, and cerebellum) is shown. Comparisons within the direction of PA show that rightward PA increases RSFC in subregions of the PPCs and between the PPCs and the right middle frontal gyrus, while left PA decreases RSFC between these regions. Both rightward and leftward PA decrease RSFC with bilateral temporal areas. In sum, right PA increases connectivity in the right fronto-parietal network, while the effects of left PA are opposite. There is evidence from another fMRI study [82] for a temporary change in the right sided Dorsal Attention Network (DAN). Specifically, the functional connectivity between the right frontal eye field and the right intraparietal sulcus of the PPC decreases immediately after PA to rightward displacing prisms, whereas functional connectivity between the right frontal eye field and the right anterior cingulate cortex increases after PA, but recovers within one hour after the end of PA. There is also evidence [83] that, after PA to a rightward shift, right inferior frontal areas in the DAN, the left anterior insula region in the Ventral Attention Network (VAN) and the right superior temporal sulcus reduce their connectivity to the left inferior parietal lobule of the PPC and the medial prefrontal cortex, which are part of the Default Mode Network (DMN). Thus, areas related to goal directed attention and action, being part of the VAN and DAN, reduce their connectivity to key nodes of the DMN, mainly involved in self-referential processing, as well as functional interactions between the ventral and the dorsal attentional networks. The results of these studies differ in several aspects, but share the finding that PA involves not only local effects in the PPC and in cerebellar regions, but impacts on a much more complex network of areas involved in spatial attentional processing. In the present case-study, there is evidence that TMA has a sub-representation of the right arcuate fasciculus for the parieto-temporal connections, a congenital-sub representation of the inferior and superior longitudinal fasciculi, a damage in the occipital dorsal lobe, in the inferior fronto-occipital fasciculus and a left < right anisotropy in the visual pathways. In the light of the neurofunctional evidence reviewed above, both the sub-representation of the right arcuate fasciculus involved in the parieto-temporal connections, and the occipital dorsal damage of the inferior fronto-occipital fasciculus (left < right anisotropy in the visual pathways) might play a role in the defective PA process, by disrupting the early recalibration phase, rather than in the successive realignment phase. Be as it may, the results of the present behavioral study results indicate that AEs may occur in the absence of a detectable recalibration phase, relying mainly on the successive spatial realignment process.

## 5. Conclusions

In sum, adaptation to prisms displacing leftward the visual scene, by a pointing procedure making use of visual, auditory, and bimodal audio-visual targets, brings about in control healthy participants proprioceptive, visuo-proprioceptive and audio-proprioceptive AEs, with the only exception of the visual straight-ahead condition with visual targets. In a patient with right hemianopia and apperceptive agnosia, AEs are, at least in part, preserved. When multisensory audio-visual targets are used for PA, all AEs occur: this complete pattern of AEs is traced back to the role of multisensory signals, which enhance the building up of AEs. The patient’s PA process features however less accurate pointing and less error correction, suggesting that, for the AEs after PA to occur, the stage of major error reduction and its feedback, of which participants are aware (“recalibration”) is less central to the building up of adaptation, as compared to the sensorimotor pointing activity per se, which supports the successive “realignment” phase. The patient’s lesion pattern, discussed with reference to data from healthy participants, indicates that cortical connections (fronto-temporal, fronto-occipital) are not crucial for AEs after PA to occur. These findings are also in line with the largely preserved occurrence of AEs in patients with a variety of cortico-subcortical lesions [20,21,22,23]. 

## Figures and Tables

**Figure 1 brainsci-12-00480-f001:**
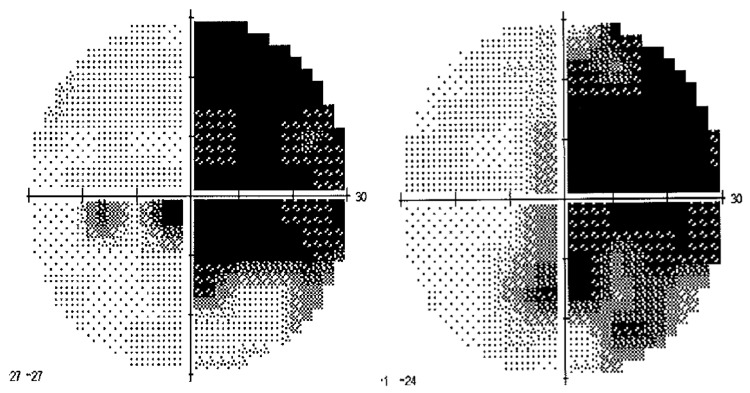
Patient TMA. Humphrey Visual Field (HVF 30.2) perimetry. Left eye: left panel; right eye: right panel.

**Figure 2 brainsci-12-00480-f002:**
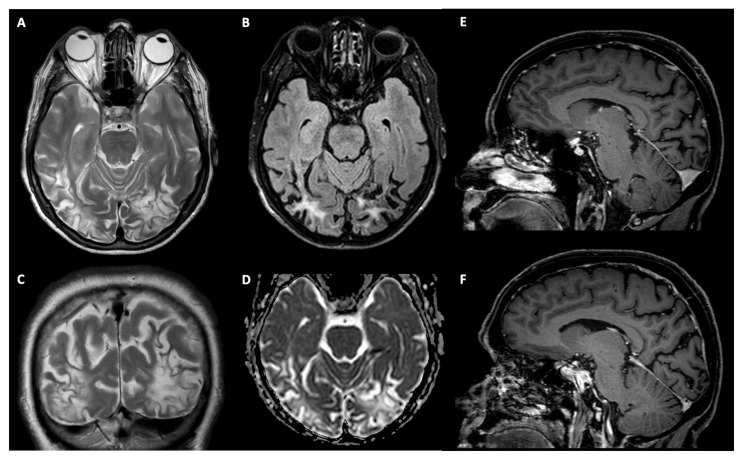
Conventional MRI. (**A**) Axial and (**C**) coronal T2-weighted images. (**B**) Axial FLAIR (Fluid-Attenuated Inversion Recovery) image. (**D**) Axial ADC (Apparent Diffusion Coefficient) map derived from DWI (Diffusion Weighted Imaging). (**E**) and (**F**) Sagittal post-gadolinium T1 images. Images show bi-occipital and right temporo-parietal cortico-subcortical gliotic-malacic sequelae with increased T2/FLAIR signal and facilitated diffusion on ADC. Bilateral atrophy of the parieto-occipital lobes is also shown, with widening of the posterior cingulate and parieto-occipital sulci ((**E**), left; (**F**), right)).

**Figure 3 brainsci-12-00480-f003:**
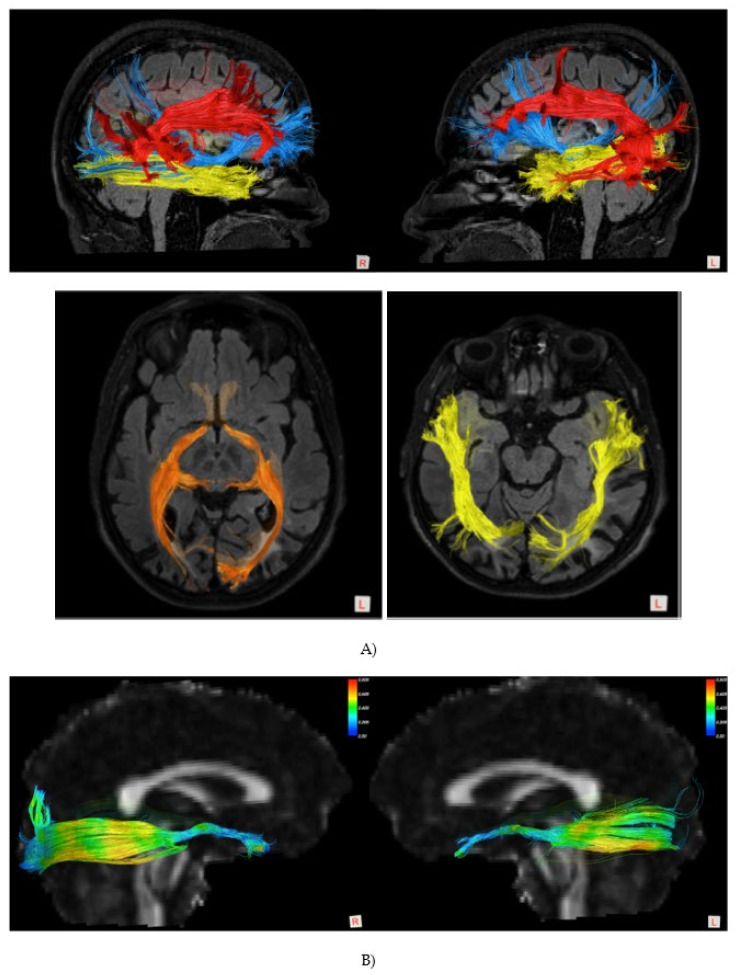
Patient TMA. (**A**) DTI imaging: RED, superior longitudinal fasciculus; ORANGE: optic pathways; YELLOW: inferior longitudinal fasciculus; BLUE: inferior fronto-occipital fasciculus (sagittal and axial images). (**B**) Fractional anisotropy: left side (L), right side (R).

**Figure 4 brainsci-12-00480-f004:**
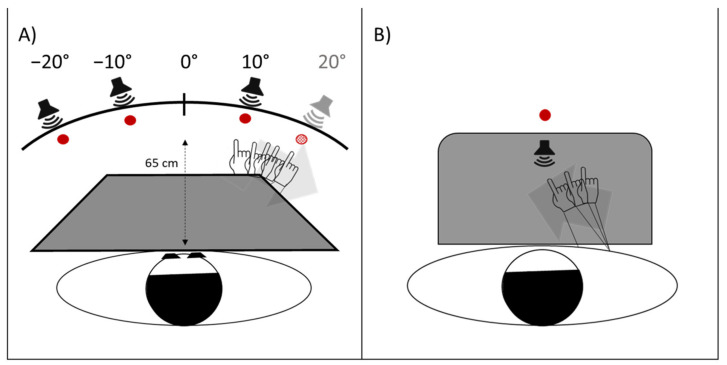
(**A**) Setup of the prism adaptation exposure phase; (**B**) setup of the pre- and post-exposure assessment of aftereffects (AEs).

**Figure 5 brainsci-12-00480-f005:**
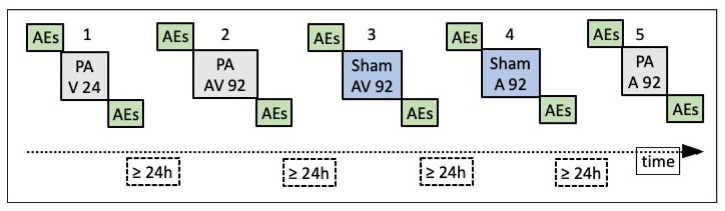
Temporal sequence of the five sessions: three real Prism Adaptation (PA) and two Sham PA sessions to visual (V), audio-visual (AV) and auditory (A) targets. Number of pointing trials for each session: 24 (session #1) or 92 (sessions #2-5). AEs were assessed in all sessions in the following order: proprioceptive (P), audio-proprioceptive (AP), visuo-proprioceptive (VP) and visual (V). The interval between one session and the following one was at least 24 h.

**Figure 6 brainsci-12-00480-f006:**
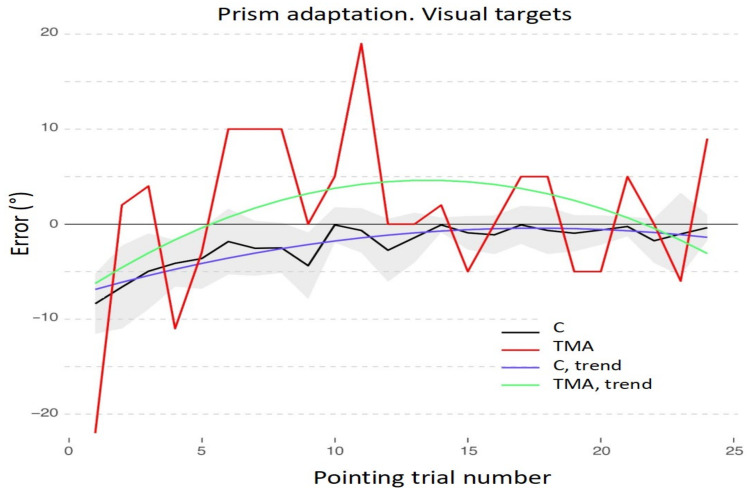
Prism adaptation with visual targets: control participants (C) and patient TMA. Trends of shift pointing errors (deg°) during the prismatic exposure phase. Mean deviations of control participants (black) and of patient TMA (red). Fitted data show the significant trends for control participants (violet line) and for patient TMA (green line). Shaded areas: ±1 SEM (Standard Error of the Mean).

**Figure 7 brainsci-12-00480-f007:**
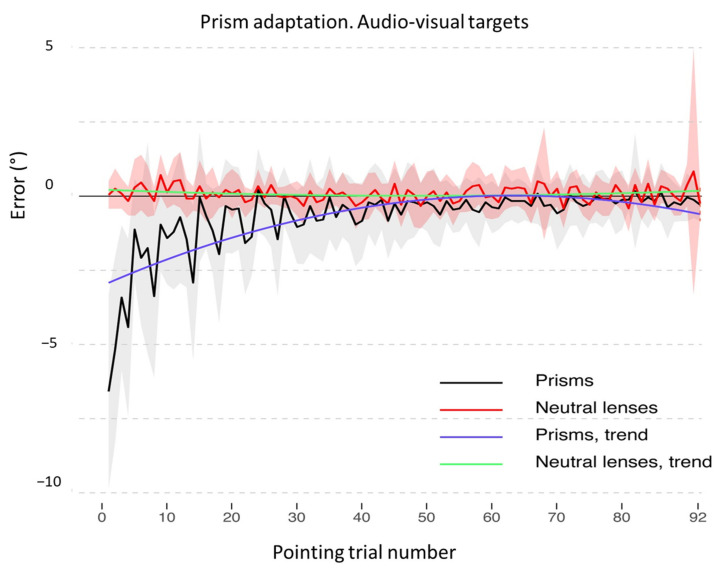
Prism adaptation with audio-visual targets: control participants. Trends of shift pointing errors (in degrees of visual angle, deg °) during the prismatic exposure phase. Mean deviations from the target with neutral (red) and with prismatic (black) lenses. Fitted data for neutral (green) and prismatic (violet) lenses show the significant trends for the two conditions. Shaded areas: ±1 SEM (Standard Error of the Mean) and deviations from the target with neutral (red) and prismatic (black) lenses. Shaded areas.: ±1 SEM.

**Figure 8 brainsci-12-00480-f008:**
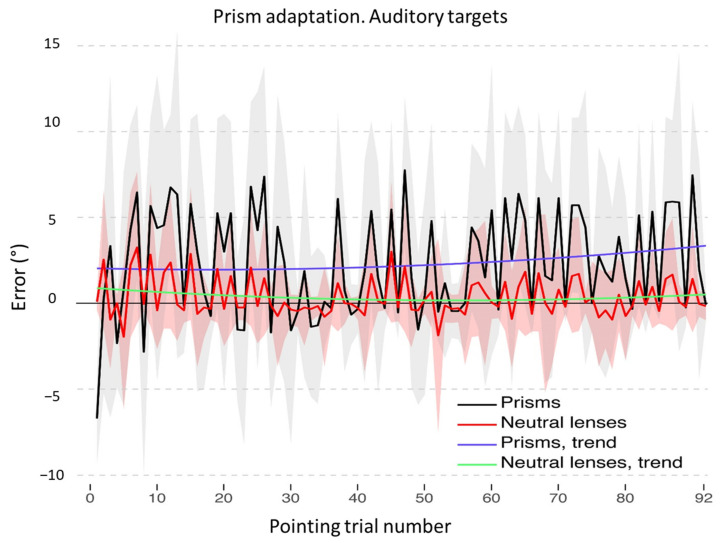
Prism adaptation with auditory targets: control participants. Trends of shift pointing errors (deg.°) during the prismatic exposure phase. Mean deviations from the target with neutral (red) and with prismatic (black) lenses. Fitted data for neutral (green) and prismatic (violet) lenses show the significant trends for the two conditions. Shaded areas: ±1 SEM.

**Figure 9 brainsci-12-00480-f009:**
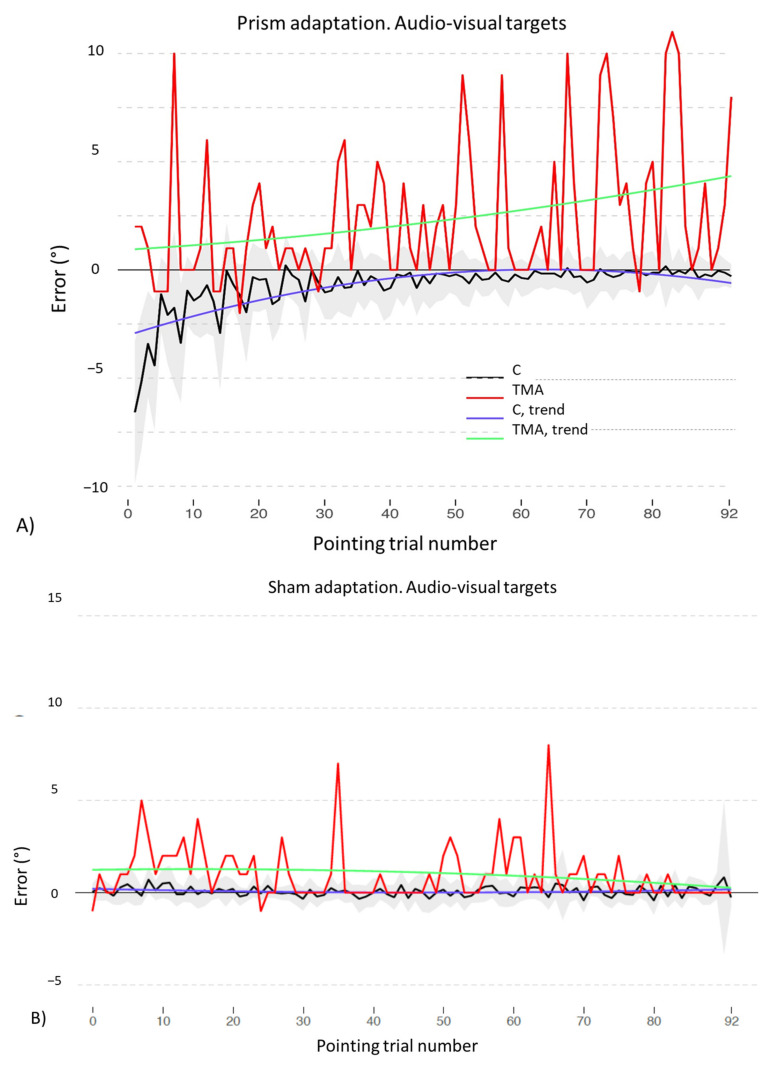
Prism adaptation (**A**) and sham adaptation (**B**) with audio-visual targets: control participants (C) and patient TMA. Trends of shift pointing errors (deg °) during the prismatic exposure phase. Mean deviations of control participants (black) and of patient TMA (red). Fitted data show the significant trends for control participants (violet) and for patient TMA (green). Shaded areas: ±1 SEM.

**Figure 10 brainsci-12-00480-f010:**
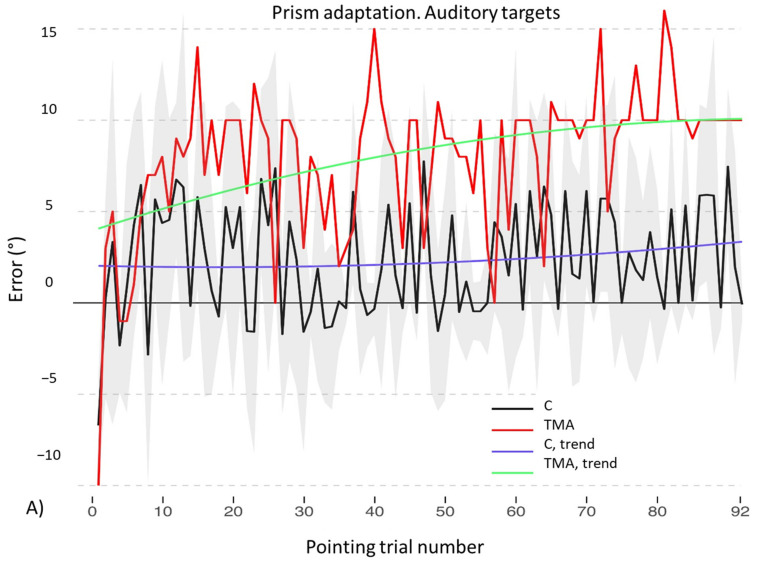
Prism adaptation (**A**) and sham adaptation (**B**) with auditory targets: control participants (C) and patient TMA. Trends of shift pointing errors (deg °) during the prismatic exposure phase. Mean deviations of control participants (black) and of patient TMA (red). Fitted data show the significant trends of control participants (violet) and of patient TMA (green). Shaded areas: ±1 SEM (Standard Error of the Mean).

**Figure 11 brainsci-12-00480-f011:**
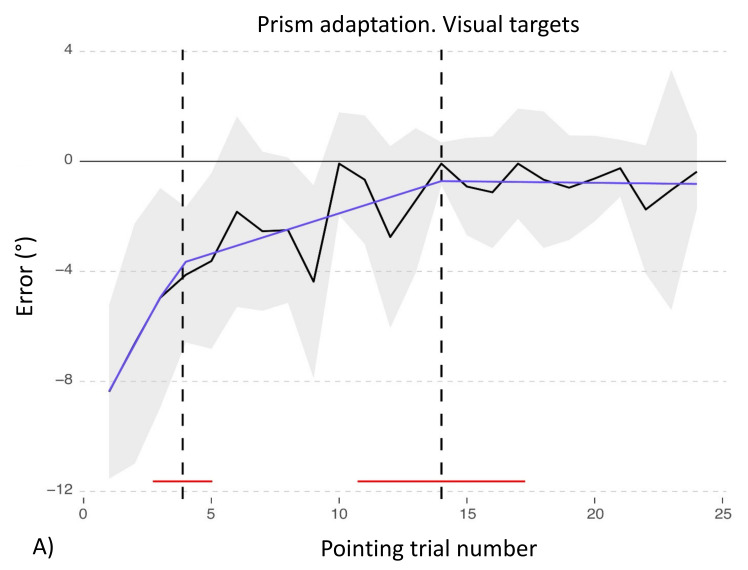
Prism adaptation in control participants. Mean deviations from the target (deg.°) at each pointing trial (black). Shaded area: ±1 SEM. Violet line: fitted data from the segmented regression. Two vertical dashed black bars: breakpoints’ estimates. Red segments: breakpoints 95% CIs. Visual targets (**A**), audio-visual targets (**B**), auditory targets (**C**).

**Figure 12 brainsci-12-00480-f012:**
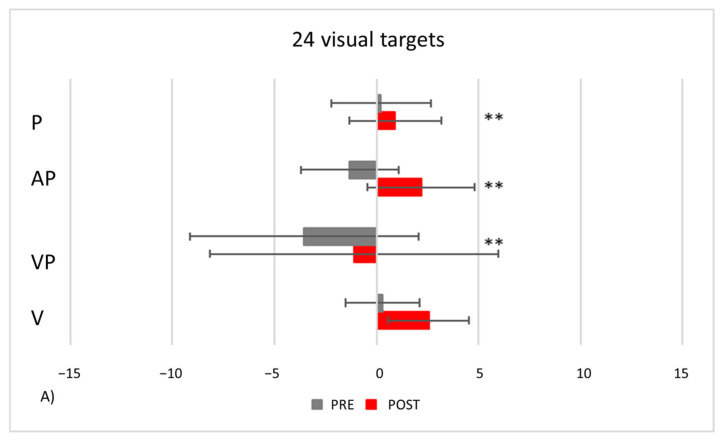
AEs in control participants. Mean (SE) deviations (deg.°) from the mid-sagittal plane of the body in the proprioceptive (P), audio-proprioceptive (AP) visuo-proprioceptive (VP) straight-ahead tests in the visual (**A**), auditory (**B**), and bimodal audio-visual (**C**) conditions of adaptation; positive values (rightward deviations from the objective body midline), negative values (leftward deviations). ** = *p* ≤ 0.001.

**Figure 13 brainsci-12-00480-f013:**
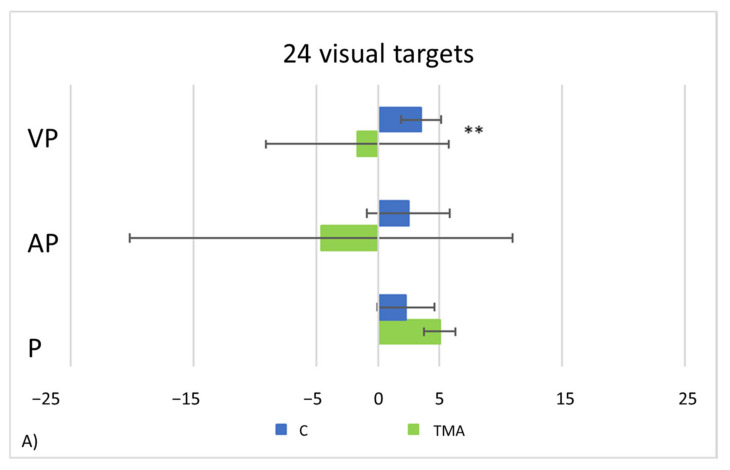
Patient TMA and control participants (C). Mean (±SE) deviations (deg.°) from the mid-sagittal plane of the body in the visuo-proprioceptive (VP), audio-proprioceptive (AP) and proprioceptive (P) straight-ahead tests in the 24 visual (**A**), 92 auditory (**B**), 92 audio-visual (**C**), 92 sham auditory (**D**) and 92 sham audio-visual target (**E**) conditions of adaptation, in control participants (C, blue) and in patient TMA (green). Positive values (rightward deviations from the objective body midline), negative values (leftward deviations). ** = *p* < 0.05.

**Figure 14 brainsci-12-00480-f014:**
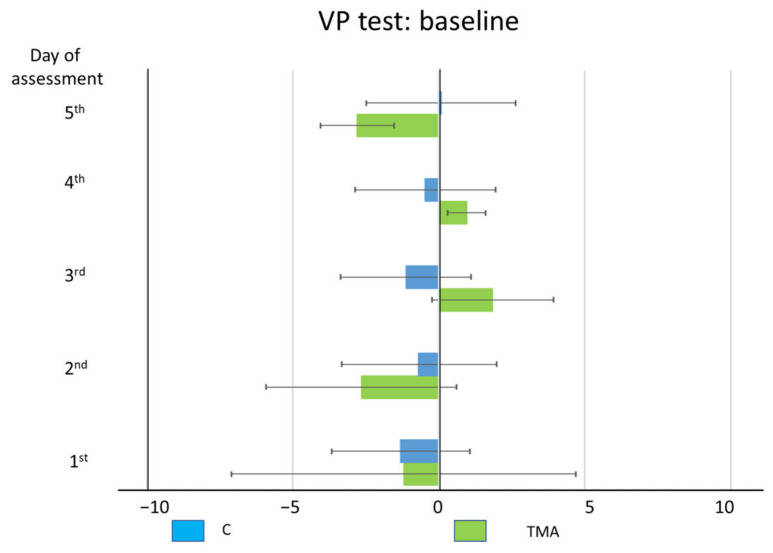
Patient TMA and control participants (C): baseline pointing performance. Mean (±SE) deviations (deg.°) from the mid-sagittal plane of the body in the visuo-proprioceptive (VP) test in the five days of assessment.

## Data Availability

Data are available at: https://mfr.osf.io/render?url=https%3A%2F%2Fosf.io%2F4sbck%2Fdownload, accessed on 6 March 2022.

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
