# Peer review of "Aftereffects to Prism Exposure without Adaptation: A Single Case Study"

_brainsci, 2022, doi:10.3390/brainsci12040480_

Round 1

Reviewer 1 Report

Overall the manuscript is detailed and presents a novel case study investigating prism adaptation across different sensory conditions - auditory, visual and A-V in a patient with bilateral atrophy in parieto-occipital areas. The patient's lesions and behavioural performance are well documented and the inclusion of auditory and A-V stimuli during prism adaptions in the present design provides novel insights. While this is overall a clear and well done manuscript that should be published, below I outline recommendations to improve the clarity and to highlight the novel nature of the findings.

1) Abstract - I found the number of acronyms made it difficult to read and understand what was done. One solution for PA would be to state in parentheses (Prism Adaption: PA) in order to clarify the meaning of PA.

2) Introduction - is overall clear and informative.

  • Around Lines 91-95, a few more sentences, some more content as it relates to what is known about the neural correlates of PA and AE is warranted given the emphasis on the neural correlates in the remainder of the manuscript.
  • Adding a few sentences with respect to dorsal-ventral visual streams (e.g., see Milner and Goodale, 2018) will contextualize the present results within the broader literature visuomotor control.
  • Line 102: "...may occur in different sensory modalities" - is not clear. Suggestion "...occur to the same extent with visual, auditory, and visual-auditory targets..."

4) Method contains the relevant details but would benefit from some additional figures and some points of clarification. Specifically:

  • A figure with a diagram of the various sessions is needed. Given the inclusion of a case study, it would make sense to me to include the order in which TMA completed the tasks.
  • Line 208 - Why only 3? Where did +20 go? The change in target numbers should be explained
  • Lines 229 - 301 - A diagram of the set-up would be helpful. In words it was difficult for me to visualize the set-up. Clarify if participants were touching the straight-ahead target 65cm away - or directing their point to the target? Again - I think a diagram will clarify any confusion for readers.
  • A table that organizes the different conditions and associated hypotheses would be helpful to organize the many comparisons going forward through the results and discussion.

5) Results are thorough. Reconsidering what is included in the main manuscript and what is included as appendices will make it easier for the readers to follow the results. Specific comments:

  • Figures 8, 9, 10, 11 should be included together as a four panel figure in order to allow readers to directly compare the four sensory conditions as this was a primary comparison in the present research.
  • I recommend moving many of the statistical findings to an appendix as reference and reporting the key findings in the text.
  • Independent variable " Number of pointings" - I recommend using "trial number"

6) Discussion is logical, but does not clearly emphasize the pattern of results across the four sensory conditions.

  • Include subheadings in discussion to help to organize what was learned from the many statistical comparisons that were made. 
  • Increase emphasis in the discussion around Auditory-Visual adaptation - AEs. On reviewing the graphs it looks as though TMAs performance in AV condition was more in line with controls, whereas Auditory and Visual led to unique AEs. Additional discussion around the AEs patterns is warranted and will better highlight the novel contribution of the current case study.

Reviewer 2 Report

This is an interesting and detailed case study of a brain-damaged patient who did not show any adaptation to prism exposure, but showed largely preserved aftereffects.

Some comments for the authors to consider:

  • Abstract, line 128 and throughout manuscript: Patient with “…bilateral atrophy of the parieto-occipital cortices”. Figure 1 rather shows the temporo-occipital lesions (only the coronal slice shows the parietal lobe) – additional slices or a lesion reconstruction would be helpful.

  • e.g. Figure 6 and 7: the patient shows an important variation in the pointing behaviour (up to 20° from one pointing to the next). I did not see any information about the neurological exam - did the patient have any ataxia (she received Cyclophosphamid and Vincristine) or optic ataxia (bilateral parietal atrophy)?

  • The patient was tested three years after she developed a leukoencephalopathy – do the authors think that the AE were always preserved or rather that they recovered due to plastic changes?
